# ENHANCING VIDEO ACTION RECOGNITION WITH VISION AND LANGUAGE COLLABORATION

## ABSTRACT

Leveraging video pre-trained models has led to significant advancements in video understanding tasks. However, due to the inherent bias towards temporal learning in video pre-training, these models fail to capture comprehensive spatial cues. Additionally, the widely-used supervised adaption methods lack fine-grained semantic guidance as single action labels cannot precisely depict the intra-class diversity. To address these challenges, we incorporate the general capabilities of large Vision Language Models (VLMs) and propose a cross-modal collaborative knowledge transfer method to enhance video understanding. First, we propose an attentive spatial knowledge transfer method that distills spatial knowledge from the VLM's image encoder, enabling the precise capture of spatial information. Next, we design a contrastive textual knowledge transfer approach that achieves detailed video representations through fine-grained text-video alignment. Owing to the cross-modal knowledge transfer, the video representations are capable of attending to informative spatial regions and aligning with fine-grained texts that carry rich semantics. Extensive experiments demonstrate that our method achieves state-of-the-art performance across various datasets, validating its effectiveness.

## 1 INTRODUCTION

Video action recognition is a fundamental task in the pursuit of intelligent video understanding. Conventional methods focus on spatio-temporal representation learning by designing exquisite backbone architectures, including convolutional neural networks (CNNs) (Arnab et al., 2021; Carreira & Zisserman, 2017) and Transformers (Fan et al., 2021; Bertasius et al., 2021; Liu et al., 2022). However, these methods typically follow a supervised learning paradigm, which comes with high costs due to the need for large-scale video data curation and labeling. Recently, there is a new research shift in improving representation modeling that relies on self-supervised techniques to learn the inherent property in videos prior to downstream adaption. By effectively utilizing uncurated video data during pre-training, these video pre-trained models exhibit powerful temporal encoding capability, facilitating various downstream tasks with remarkable performance (Tong et al., 2022; Feichtenhofer et al., 2022a; Huang et al., 2023; Zhao et al., 2024b; Li et al., 2023a). Despite these advancements, there are still some limitations. Firstly, video pre-trained models often lack sufficient spatial understanding due to their video-centric pre-training objectives, making their direct adaptation to downstream tasks suboptimal especially when actions differ in subtle spatial cues (as illustrated in Fig. 1 (a), those actions such as '*rock scissor paper*' and '*dunking basketball*' require precise spatial cues for identification). To enhance the spatial understanding, some methods (Wang et al., 2024; Zhao et al., 2024a) scale the video pre-training dataset to an extremely large size. Although achieving excellent results, they suffer from significant training costs (*e.g.* (Wang et al., 2024) needs over 8,000 A100 GPU days). While the other line enjoys the strong spatial encoding ability of an off-the-shelf image pre-trained model and adapt it to video tasks (Pan et al., 2022; Yang et al., 2023). Recently, with the success of vision-language models (VLM), transferring from VLM has become a more effective approach to enhance spatial representation learning (Lin et al., 2022; Qing et al., 2023; Chen et al., 2024; Zhang et al., 2024). Despite the expressive results, they require meticulously designed temporal fusion modules, yet solely training these modules on video downstream datasets is still weak at achieving comprehensive temporal understanding (Lee et al., 2024).

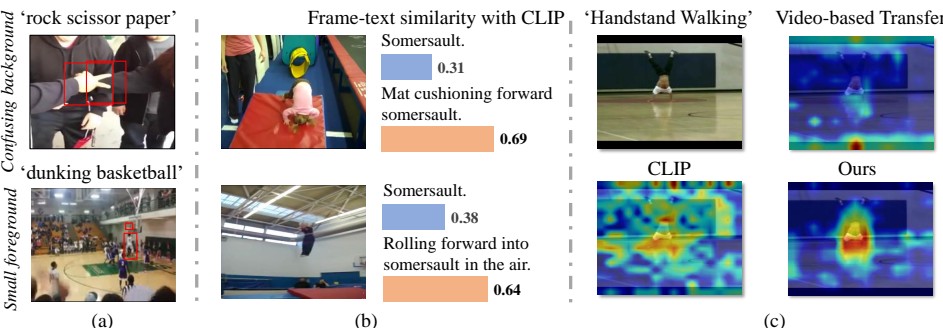

Figure 1: (a) Illustration of the actions that need subtle spatial cues to identify. It shows the model must either distinguish the action subject from a confusing background or localize the small but critical area where the action occurs, both of which demand precise spatial cues. (b) Illustration of the limitation of single-label supervision. We observe that examples are semantically closer to different descriptions rather than the single action name, highlighting the incompleteness of using a single action label. (c) Illustration of the spatial capture ability of different models. The example shows the video-based transfer is weak at fully focusing on the foreground, while VLM (*e.g.* CLIP) is capable of encoding the critical area but prefers a broader attention pattern, and our model utilize such advantage with minor revision that leads to a precise spatial representation.

Additionally, transfer learning methods typically adopt a supervised fine-tuning strategy to adapt the pre-trained model to downstream tasks. However, these approaches often rely on a single label for supervision, providing limited semantic guidance, particularly in video tasks (as illustrated in Fig. 1(b), although the samples belong to the same action '*somersault*', they exhibit various action forms and occur in diverse scenarios, making it insufficient for a single action label to capture the full semantics of all samples.). Training with such supervision indeed leads to the incomplete representation. Recently, with the strong text encoding capabilities of VLM, some VLM-based transfer methods have begun using action names or fine-grained descriptions as stronger supervision signals. They either use single description (Qing et al., 2023; Ahmad et al., 2023; Huang et al., 2024) or average multiple description at the text space (Zhang et al., 2024). However, the single text feature is still incomplete to describe all possible scenarios where and how an action occurs.

Based on the investigation above, we attribute these issues to a lack of spatial and semantic understanding. To address these, we propose a cross-modal collaborative knowledge transfer method based on the video pre-trained model, leveraging VLM to enhance spatial and semantic learning while preserving the temporal understanding capabilities of the video pre-trained model. Firstly, we leverage the powerful capabilities of VLM in encoding spatial details (illustrated in Fig. 1 (c)) to enhance the spatial understanding of video pre-trained models. Specifically, we propose an attentive spatial knowledge transfer module to enhance the generalization of video features through feature distillation. To address the domain gap between VLM and video features—where VLM often captures broader content, inevitably including irrelevant information to confuse video classification—we design a gating mechanism that filters out unnecessary information, making the distillation process more focused and effective. Secondly, to comprehensively understand the actions beyond class labels, we propose a contrastive textual knowledge transfer module to learn the general semantic knowledge based on the text-side of VLM. Apart from the previous methods that utilize single action name or single description (Qing et al., 2023; Ahmad et al., 2023; Huang et al., 2024; Chen et al., 2024), we propose a decompose-expand prompting method to create a description bank that stores fine-grained and diverse description candidates for each action. To fully harness the semantics of these descriptions, we introduce a visual-guided assignment method to generate customized supervision signals for each sample. Moreover, a cross-modal contrastive loss is deployed to guide the video pre-trained model to learn the complete action concept.

We summarize the contributions as follows:

- We offer a novel perspective of collaborating VLM and video pre-trained model for video understanding, leveraging their complementary abilities to achieve the comprehensive video representation.

- To fully harness the potential of VLM, we propose an attentive spatial and constrastive textual knowledge transfer method to augment the spatial and semantic learning during transfer learning.
- Experiments on various downstream tasks demonstrate the superiority of our method.

## 2 RELATED WORK

**Video Action Recognition.** Conventional methods focus on spatio-temporal learning under fully-supervised settings, where all categories are predefined. These approaches have achieved remarkable performance using various architectures, including convolution networks (Carreira & Zisserman, 2017; Feichtenhofer et al., 2019) and transformers (Arnab et al., 2021; Bertasius et al., 2021; Fan et al., 2021). In addition to the architecture design, self-supervised video representation learning (Diba et al., 2021; Feichtenhofer et al., 2019; 2022a; Jenni & Jin, 2021; Zhao et al., 2024b) has also gained popularity. Recently, more and more works focus on the zero-shot setting (Lin et al., 2023; Huang et al., 2024), which aims constructing a model with strong generalization ability to recognize unseen categories. These models often utilize large vision-language model to help align text-video feature so as to easily retrieve the unseen categories by their texts. In this work, we utilize both video and image-language model to achieve considerable performance under both settings.

**Vision-Language Model.** In recent years, vision-language models (VLM) (Chen et al., 2022b; Jia et al., 2021; Li et al., 2023b) have made remarkable progress. One of the most remarkable and influential works is CLIP (Radford et al., 2021), which is trained on 400M data following a contrastive manner, and shows remarkable performance on zero-shot image classification. The success of VLM inspires the "fine-tuning" trend on multiple downstream tasks, such as open-vocabulary detection (Gu et al., 2021), segmentation (Liang et al., 2023), caption (Mokady et al., 2021). In the video-related tasks, it also has been explored to various tasks, such as video action recognition (Chen et al., 2024; Zhang et al., 2024), action localization (Ju et al., 2023). Previous works usually treat the VLM as part of backbones. In this work, we utilize the VLM as a powerful book that absorbs its knowledge but doesn't apply it as our backbones.

**Transfer Learning.** Transferring useful knowledge from large foundation models has achieved remarkable performance for various downstream tasks, including video action recognition (VAR). Conventional transfer learning for VAR utilizes video pre-trained model by employing fine-tuning or linear-probe (Kumar et al., 2022; Feichtenhofer et al., 2022a; Zhao et al., 2024b). Besides, knowledge distillation is another effective way for transfer learning, such as logits and feature distillation (Yang et al., 2024). Recently, adapting VLM to VAR has achieved outstanding performance. Some of them mainly revise the image encoding mechanism by inserting exquisite temporal-fusion module to transfer knowledge from image domain (Chen et al., 2022a; Yang et al., 2023; Pan et al., 2022), while others dedicate to align text-video feature space to best leverage the language supervision ability of VLM. In addition to simply using action names as text supervision (Qing et al., 2023; Ni et al., 2022), (Chen et al., 2024) and (Zhang et al., 2024) employ LLM to generate fine-grained descriptions for each action. In this paper, we construct an effective transfer mechanism to incorporate video pre-trained model and VLM to VAR task.

## 3 METHOD

As illustrated in Fig. 2, our framework is composed of three key components: (1) **Video Adaptation** to transfer the spatio-temporal information from a video pre-trained model. (2) **Attentive Spatial Knowledge Transfer** to help obtain more generalized spatial feature with the aid of the spatial knowledge of VLM. (3) **Contrastive Textual Knowledge Transfer** to provide fine-grained language supervision based on semantic knowledge of VLM.

### 3.1 VIDEO ADAPTATION

Video adaptation aims to adapt a video pre-trained model to downstream task, which is typically performed in a fully-supervised way. Specifically, given a video clip $\mathbf{X}_v \in \mathbb{R}^{T \times H \times W \times 3}$ ($T$, $H$ and $W$ represent the frame number, height, and width, respectively). Following UMT (Li et al., 2023a), each frame is divided into $K = \frac{H}{P} \times \frac{W}{P}$ patches, and the size of each patch is denoted as

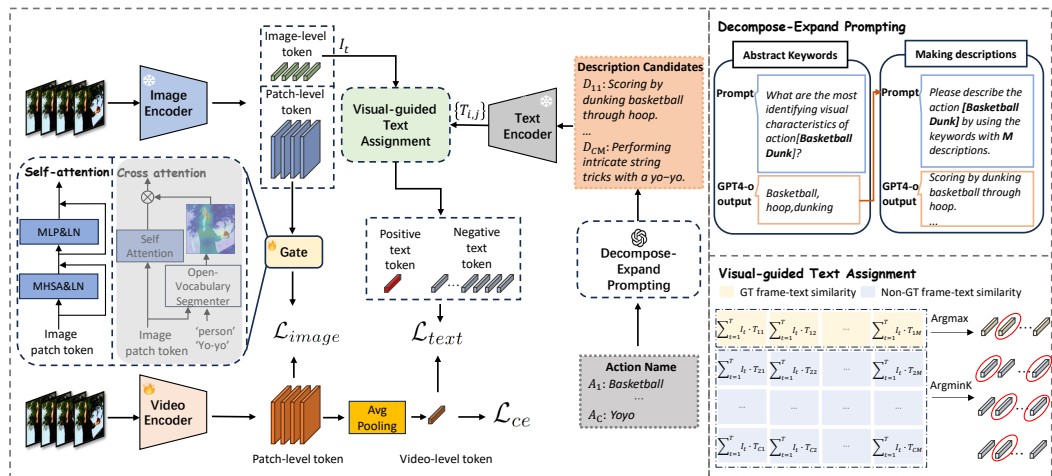

Figure 2: The overall framework of our method. It consists of three key components: The **video adaptation** uses cross-entropy loss on a trainable video encoder to create temporal-sensitive features. The **attentive spatial knowledge transfer** component integrates spatial information from the image encoder through gated distillation, for which we propose two types of gating mechanisms to enhance exploration. The **contrastive textual knowledge transfer** component leverages cross-modal contrastive loss with a large set of precise and diverse description candidates.

$P \times P$. These patches are then projected by 3D convolution. Thus the projected embeddings can be formulated as $\mathbf{Z}_v^{(0)} = \left\{ z_{v,i}^{(0)} \right\}_{i=1}^{N}$, where $N = T \times K$ represents the total patch number within a video clip. Assuming that the video encoder has $L$ transformer blocks, the embeddings of the $l$-th layer can be extracted by:

$$\mathbf{Z}_v^{(l)} = \text{Video-Block}^{(l)}(\mathbf{Z}_v^{l-1}) \in \mathbb{R}^{T \times K \times D}, \tag{1}$$

where $D$ refers to the feature dimension. Notably, the spatio-temporal attention mechanism is used to encourage the interactions among all the tokens.

To transfer the spatio-temporal information to the downstream datasets, the patch-level tokens $\mathbf{Z}_v^{(L)}$ from the last layer are first converted to the video-level token $\mathbf{Z}_v^{(cls)} \in \mathbb{R}^D$ by average pooling at the spatial dimensions, followed by a linear projection layer $\Phi$ to obtain the classification logits. Finally the cross entropy loss is adopted for training. The process can be formulated as:

$$p_v = \Phi\left(\mathbf{Z}_v^{cls}\right) \in \mathbb{R}^C, \tag{2}$$

$$p_{v,j} = \frac{\exp(p_{v,j})}{\sum_{j'=1}^{C} \exp(p_{v,j'})}, \tag{3}$$

$$\mathcal{L}_{ce} = -\sum_{j=1}^{C} y_j \log(p_{v,j}), \tag{4}$$

where $C$ represents the number of the class and $y_j \in \{0, 1\}$ denotes the ground-truth label.

## 3.2 ATTENTIVE SPATIAL KNOWLEDGE TRANSFER

To guide the video features to attend to the informative spatial regions, we inject the spatial knowledge from VLM to the video encoder. As VLM is trained by massive image-text pairs (*e.g.*, 400M for CLIP), the image encoder of VLM comprehends a wide range of visual concepts and exhibits advantages in capturing spatial details. Previous works (Yang et al., 2023; Pan et al., 2022; Lin et al., 2022) tend to directly adapt the image encoder to video tasks by equipping with various temporal-interaction modules. However, these approaches are insufficient for fully exploring spatio-temporal learning. Hence, we opt to freeze the image encoder to keep the generalization abilities and treat it

as an external knowledge resource. Based on the observation that the last layer obtains high-level spatial information (Hajimiri et al., 2024), we propose a distillation approach to transfer the spatial knowledge on the last layer.

Specifically, given a video clip $\mathbf{X}_v$, we use the video encoder to obtain the video patch features $\mathbf{Z}_v^L \in \mathbb{R}^{T \times K \times D}$, and use image encoder to obtain the frame patch features $\mathbf{Z}_m^L \in \mathbb{R}^{T \times K' \times D'}$. Considering the video features and frame features has different number of patches we firstly interpolate the video feature up to the same patch number as the image feature, which is denoted as $\mathbf{Z}_{\hat{v}}^L \in \mathbb{R}^{T \times K' \times D}$. A straightforward way is to use feature distillation between $\mathbf{Z}_{\hat{v}}^L$ and $\mathbf{Z}_m^L$, but we find such approach fail to largely boost the performance, which is attributed to the domain gap between video and image (Huang et al., 2024). Thus, we propose a gated feature distillation method. The motivation is that the image encoder contains sufficient spatial information but not all of them are necessary for video action recognition. For example, it is required to encode the spatial information about '*hands*','*ball*','*hoop*' for the action '*basketball dunk*', but the image encoder of VLM will focus on all the object information such as '*sky*' and '*ground*'.

**Self-Attention**. *Attention layer* (Vaswani, 2017) plays the role of gate in self-Attention uses *attention layer* (Vaswani, 2017), where the spatial information is extracted by the interaction within the image patch tokens, formulated as:

$$\mathbf{G}_m = \text{Softmax}(\frac{\mathbf{Q}\mathbf{K}^T}{\sqrt{D}})\mathbf{V}, \qquad \mathbf{Z}_m^g = \text{MLP}\left(\text{LN}\left(\mathbf{G}_m\right)\right) + \mathbf{G}_m, \tag{5}$$

where $\mathbf{Q}$, $\mathbf{K}$, $\mathbf{V}$ refer to the image patch tokens $\mathbf{Z}_m^L$ that after the linear projection layer. Then, we use L1 loss to narrow the gap between video and image tokens: $\mathcal{L}_{image} = \left|\mathbf{Z}_m^g - \mathbf{Z}_{\hat{v}}^L\right|$.

**Cross-Attention**. This gate mechanism utilizes the extra text information for precise distillation. To be specific, the ideal spatial information is located in the foreground area. As we has gotten the keywords for each action from GPT-4o, we utilize an open-vocabulary semantic Segmentation model (e.g. NACLIP (Hajimiri et al., 2024)) to excavate the area that is corresponding to the keywords. Then merging them as the foreground area. The process can be formulated as:

$$\mathbf{A}_{t,w} = \text{NACLIP}(T_w, \mathbf{X}_{v,t}), w = 1, 2, ..W \tag{6}$$

$$R_{t,w} = \left(\mathbf{Z}_{m,t}^{cls} \cdot T_w\right) \tag{7}$$

$$\tilde{A}_{t,w} = \begin{cases} 1 & \text{if } \mathbf{A}_{t,w} >= thr1 \text{ and } R_{t,w} > thr2 \\ 0 & \text{otherwise} \end{cases}, \tag{8}$$

where $\mathbf{X}_{v,t}$ represents $t$-th frame, $\mathbf{Z}_{m,t}^{cls}$ represents frame-level embedding for the $t$-th frame, $T_w$ represents the keyword text, $w$ represents the number of keywords and $\mathbf{A}_{t,w}$ represents an attention map reflecting the key area. We use max-pooling operation alone the $w$ dimension on $\tilde{A}_{t,w}$ to get the overall foreground mask $\tilde{A}_t$ for each frame, and concat the mask as $\tilde{A}$. Notably, here we use a image-guided strategy to filter out the attention map of the keyword that may not exist. Finally, we apply the L1 loss between the video feature and the gated image feature: $\mathcal{L}_{image} = \left|\mathbf{Z}_m^g - \mathbf{Z}_{\hat{v}}^L \otimes \tilde{A}_t\right|$.

In the ablation study, we experiment the two gate strategies and find the self-attention gate mechanism achieves slightly higher performance than the cross-attention gate mechanism. We conjecture that the current open-vocabulary semantic segmentation model performs badly due to the large domain shifts between image and video datasets.

## 3.3 CONTRASTIVE TEXTUAL KNOWLEDGE TRANSFER

In cases where actions involve the large intra-variance, relying solely on cross-entropy loss makes it challenging to capture this variance, leading to suboptimal adaption. Motivated by the fact that the text component of VLM enables to encode fine-grained semantic knowledge learned from the pre-training on large-scale image-text pairs, we aim to transfer this advantage to guide the comprehensive learning of video semantics. To the end, we first introduce a decompose-expand prompting method that generates precise and diverse descriptions for each class beyond the action label. Subsequently, we explore a visual-guided text assignment method to select high-quality positive and negative texts, which are then utilized in cross-modal contrastive loss.

### 3.3.1 DECOMPOSE-EXPAND PROMPTING

To depict the large intra-variance in video, we aim to provide fine-grained and diverse descriptions for each class. To inspire the comprehensive text generation ability in large language model (LLM), we design a decompose-expand prompting approach. The motivation is that each action could be depicted in different ways but should contain some certain key words.

We first query a large language model (*e.g.*, GPT-4o) to extract the key words for each action, then we ask the language model to generate the diverse descriptions based on these keywords. Please refer to appendix A.2 for more detail.

In this way, given a dataset of $C$ categories, we obtain $C \times M$ descriptions. Besides, we ask GPT-4o to provide a longer description. Then we pass all the descriptions into the text encoder and obtain their text embeddings $\mathbb{T}_{des} = \left\{ T_i \in \mathbb{R}^{M \times D} \mid i = 1, 2, ..C \right\}$ for each class.

### 3.3.2 CROSS-MODAL CONTRASTIVE LOSS

Base on the diverse descriptions for each action, we exploit them for text-video alignment through a cross-modal contrastive loss. The intuition is that a video should be matched to the most similar description and be far way from the dissimilar descriptions. In order to mine the descriptions that meet the conditions, we propose a visual-guided assignment strategy. Specifically, given a video clip $\mathbf{X}_v$, we firstly obtain the frame-level embeddings of all frames through the image encoder of VLM. Assuming the image encoder turns each frame into $K'$ patches and has $L'$ transformer blocks, we could get $T$ CLS tokens in total, which can be expressed as:

$$\mathbf{Z}_{m,t}^{(l)} = \text{Image-Block}^{(l)} \left( \mathbf{Z}_{m,t}^{(l-1)} \right) \in \mathbb{R}^{K' \times D}. \tag{9}$$

$$\mathbf{Z}_{m,t}^{cls} = \mathbf{Z}_{m,t,0}^{(L')}. \tag{10}$$

We then compute the cosine similarity between frame-level embeddings $\mathbf{Z}_{m,t}^{cls}$ and all text embeddings $\mathbb{T}_{des}$. To select the conceptually correct positive description, we consider the union of generated descriptions and the original ground-truth action names as text candidates. From these, we choose the one with the highest image-text similarity score as positive text defined as $\mathbb{T}_{pos}$. This strategy ensures that the selected positive description has a similarity score equal to or higher than that of the original action name. For creating negative descriptions, we select the bottom-k candidates with the lowest similarity scores as $\mathbb{T}_{neg}$, excluding the ground-truth descriptions. This ensures that the selected negative descriptions are irrelevant to the training sample.

Finally, we utilize a cross-modal classification loss for fine-grained text-video alignment.

$$\mathbf{Z}_v^m = \Phi_m(\mathbf{Z}_v^{cls}) \tag{11}$$

$$\mathcal{L}_{text} = -\log \frac{\sum_{T_{i'} \in \mathbb{T}_{pos}} \exp(\mathbf{Z}_v^m \cdot T_{i'}/\tau)}{\sum_{T_{i'} \in \mathbb{T}_{pos} \cup \mathbb{T}_{neg}} \exp.(\mathbf{Z}_v^m \cdot T_{i'}/\tau)} \tag{12}$$

### 3.4 TRAINING AND INFERENCE

During the training stage, the overall framework is trained based on three losses: $\mathcal{L} = \mathcal{L}_{ce} + \mathcal{L}_{text} + \alpha \cdot \mathcal{L}_{image}$. where $\alpha$ is a hyper-parameter. During the inference stage, for fully-supervised experiments, we directly use Eq. 2 as the output logits, and for zero-shot experiments, we use the similarity score between the video feature and action descriptions as the output logits.

## 4 EXPERIMENTS

**Datasets**. Our proposed method is evaluated on three widely used video action recognition datasets: Kinetics-400 (Carreira & Zisserman, 2017), HMDB51 (Kuehne et al., 2011), UCF101 (Soomro et al., 2012). Kinetics-400 consists of approximately 240k training and 20k validation videos, covering 400 classes, with each clip spanning around 10 seconds. UCF101 contains 13,320 video clips with 101 classes, and HMDB51 consists of 7000 videos with 51 classes.

Table 1: Comparison with the state-of-the-art methods on Kinetics-400. We here report the volume of the parameter that is used during inference. **Bold** indicates the best result, and underline represents the second-best result.

| Method | Backbone | Frames | FLOPs/View(T) | Views | Params(M) | Top-1 |
|---|---|---|---|---|---|---|
| *Video based transfer* | | | | | | |
| SIFA (Long et al., 2022) | Swin-B | 64 | 0.27 | 3×4 | - | 83.1 |
| ST-MAE (Feichtenhofer et al., 2022b) | ViT-B | 16 | 0.18 | 3×7 | 87 | 83.3 |
| VideoMAE (Tong et al., 2022) | ViT-B | 16 | 0.18 | 3×5 | 87 | 81.5 |
| MGMAE (Huang et al., 2023) | ViT-B | 16 | 0.18 | 3×2 | 87 | 81.7 |
| AMD (Zhao et al., 2024b) | ViT-B | 16 | 0.18 | 3×5 | 87 | 82.2 |
| *VLM-based transfer* | | | | | | |
| AIM (Yang et al., 2023) | ViT-B | 16 | 0.40 | 1×3 | 97 | 84.5 |
| DiST (Qing et al., 2023) | ViT-B | 16 | 0.32 | 1×3 | 112 | 84.4 |
| ALT (Chen et al., 2024) | ViT-B | 16 | 0.22 | 1×3 | 134 | 84.8 |
| MoTED (Zhang et al., 2024) | ViT-B | 16 | 0.34 | 1×3 | 116 | 85.4 |
| Qian et al. (2024) | ViT-B | 16 | 0.19 | 1×3 | 93 | 86.1 |
| AIM (Yang et al., 2023) | ViT-B | 32 | 0.81 | 1×3 | 97 | 84.7 |
| DiST (Qing et al., 2023) | ViT-B | 32 | 0.65 | 1×3 | 112 | 85.0 |
| ALT (Chen et al., 2024) | ViT-B | 32 | 0.33 | 1×3 | 134 | 85.5 |
| MoTED (Zhang et al., 2024) | ViT-B | 32 | 0.68 | 1×3 | 116 | 86.2 |
| Qian et al. (2024) | ViT-B | 32 | 0.38 | 1×3 | 93 | 86.2 |
| Ours | ViT-B | 16 | 0.46 | 1×3 | 87 | 86.3 |
| Ours | ViT-B | 32 | 1.10 | 1×3 | 87 | **86.9** |

**Implementation Details**. We employ UMT-B (Li et al., 2023a) as the video pre-trained model, which utilizes ViT-B as its backbone. For VLM model, we choose CLIP-L that is pre-trained by CLIP-400M as our text and image encoder. We follow the same training settings as (Li et al., 2023a). For Kinetics-400, we adopt an AdamW optimizer with the learning rate of $1.5 \times 10^{-4}$. The network is trained with 40 epochs (including a five-epoch warmup) and a weight decay of 0.05. We conduct all experiments on with 7 NVIDIA 3090 GPUs. We empirically set $\alpha$ as 10 and $\tau$ as 0.07. We achieve the best results in case of selecting one positive text for both Kinetics-400 and UCF101, and 3,500 and 1,200 negative texts for Kinetics-400 and UCF101, respectively.

Table 2: Generalization Evaluation of our method.

| Video pre-trained model | Dataset | Method | Top-1 |
|---|---|---|---|
| VideoMAE (Tong et al., 2022) | UCF101 | Baseline | 90.30 |
| | | Ours | **91.67** |
| AMD (Zhao et al., 2024b) | UCF101 | Baseline | 97.14 |
| | | Ours | **98.09** |

| Video pre-trained model | Dataset | Method | Top-1 |
|---|---|---|---|
| UMT (Li et al., 2023a) | K400 | Baseline | 84.82 |
| | | Ours | **86.30** |
| UMT (Li et al., 2023a) | UCF101 | Baseline | 95.37 |
| | | Ours | **96.93** |

(a) Performance evaluation on varied backbones.  (b) Performance evaluation on varied datasets.

## 4.1 FULLY SUPERVISED COMPARISON

**Results on Kinetics-400**. In Table 1, we compare our method with state-of-the-art approaches. It outperforms both video-based and VLM-based transfer methods across 16 and 32 frame settings. Specifically, compared to video-based transfer methods, our approach surpasses the SOTA by 4.1%. We attribute this improvement to the semantic and spatial enhancements provided by VLM. When compared to VLM-based methods, our approach outperforms the SOTA by 0.7%. Notably, our approach with 16 frames surpass all VLM-based methods utilizing 32 frames, underscoring the superiority of our approach, which incorporate the complementary ability of video pre-trained model and VLM . In addition, to clearly demonstrate the effectiveness of our method, we compare it with the baseline in Table 2b, which uses the same pre-trained model but is trained solely with $\mathcal{L}_{text}$. Our method outperforms the baseline by 1.48%, validating the VLM-guided enhancement for video understanding. Moreover, thanks to our distillation strategy, our method achieves relatively high classification accuracy with fewer parameters for inference compared to VLM transfer methods.

## 4.2 GENERALIZATION EVALUATION

As our method enhances the action recognition performance without any restrictions on video pre-trained models or video datasets, we apply our method to various backbones and datasets to verify the generalization ability of our method. As is shown in Table 2a, we first apply our approach to different video pre-trained models. We test our approach on VideoMAE (Tong et al., 2022) and AMD (Zhao et al., 2024b) on UCF101. We set the transfer training with only $\mathcal{L}_{ce}$ as the baseline. It can be seen that our approach boosts the VideoMAE baseline by 1.37% and the AMD baseline by 0.95%. We then test our approach on different datasets: K400 and UCF101 under the same backbone in Table 2b. Our method consistently achieves significant improvements across both datasets, with a particularly notable 1.48% boost on the large K400 dataset, highlighting the strong enhancements provided by the vision and language integration from VLM.

## 4.3 ABLATION

We conduct ablation study on UCF101 with 8 frames to verify our proposed method. We utilize video pre-trained model UMT-B (Li et al., 2023a) trained with only $\mathcal{L}_{ce}$ as our baseline.

**Contributions of each component.** Our method mainly contains two auxiliary components to enhance video understanding: Contrastive Textual Knowledge Transfer and Attentive Spatial Knowledge Transfer. To clearly asses the contribution of each component, we provide comprehensive experiments of all possible combinations. As shown in Table 3, solely using $\mathcal{L}_{text}$ achieves a Top-1 accuracy of 95.82%, that outperforms the baseline of 95.37% Top-1 accuracy, demonstrating the effectiveness of the cross-modal contrastive loss. Besides, incorporating both the conventional classification loss $\mathcal{L}_{ce}$ and fine-cross-modal contrastive loss $\mathcal{L}_{text}$ further boosts the accuracy to 96.56%, indicating a semantic enhancement. Combining $\mathcal{L}_{ce}$ and $\mathcal{L}_{image}$ (the fourth row) could also signif-

Table 3: Contributions of different components.

| $\mathcal{L}_{ce}$ | $\mathcal{L}_{text}$ | $\mathcal{L}_{image}$ | Top-1 |
|---|---|---|---|
| ✓ | | | 95.37 |
| | ✓ | | 95.82 |
| ✓ | ✓ | | 96.56 |
| ✓ | | ✓ | 96.27 |
| | ✓ | ✓ | 96.06 |
| ✓ | ✓ | ✓ | **96.93** |

icantly improve classification accuracy over the baseline with only $\mathcal{L}_{ce}$. Interestingly, we found that simply using $\mathcal{L}_{text}$ and $\mathcal{L}_{image}$ leads to fewer improvements on accuracy, indicating these losses are not orthogonal. The best performance is observed when all three losses are combined, achieving a Top-1 accuracy of 96.93%. This indicates that leveraging both textual and spatial information in conjunction with the cross-entropy loss significantly enhances the model's performance.

**Number of negative Texts**. The proposed Contrastive Textual Knowledge Transfer branch includes a hyper-parameter $k$ that controls the number of negative texts. We conducted experiments using all three losses, varying only $k$. Besides, we run the experimens with multiple randomseeds to obtain stable results. As shown in Fig. 3, The box plot reveals slight fluctuations when k < 1000 and a noticeable increase as k > 1000, which can be attributed to the inclusion of more hard negative descriptions. Based on these results, we select the optimal k = 1200 for all experiments.

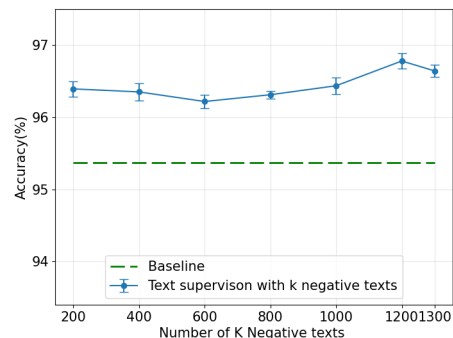

Figure 3: Ablation on the number of negative text selected for $\mathcal{L}_{text}$.

**Gate Mechanisms**. We perform an ablation study on three types of gate mechanisms: Identity, self-attention, and cross-attention. The Identity mechanism directly uses image patch features in $\mathcal{L}_{image}$, while the other two methods represent the mechanisms described in the Method section. Without any gate mechanism, we achieve a Top-1 accuracy of 96.56%. The cross-attention mechanism improved accuracy to 96.80%. The introduction of self-attention further enhances Top-1 accuracy to 96.93%, matching the best performance. Visual-

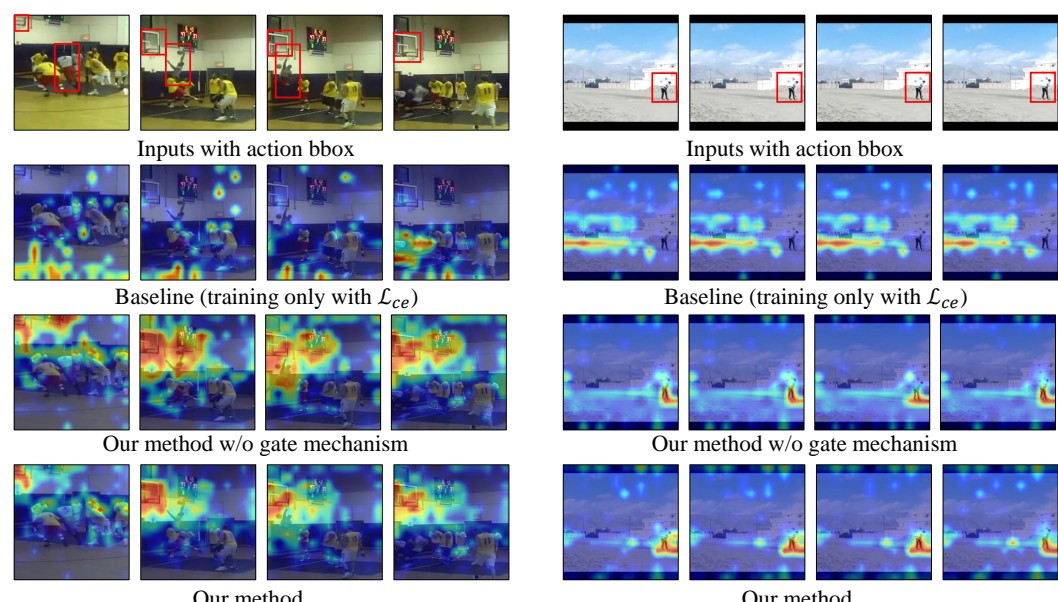

Figure 4: Left: Visualization of attentive area of the action "Dunking basketball". Right: Visualization of attentive area of the action "Cricket Bowling". We use bounding boxes to highlight the areas where actions occur. It can be seen that our method shows superior ability in capturing the critical areas through semantic and visual enhancement.

izations in Fig. 4 show that the self-attention method effectively reduces background distractions, highlighting the necessity of the gate mechanism.

Additionally, the performance of self-attention slightly surpasses that of cross-attention. To explore the cause of this discrepancy, additional experiments are conducted in Appendix A.2. Appendix A.2 reveals that the suboptimal performance of the Cross-Attention method arises from unstable segmentation results, particularly when the objects are either too small or obscured.

**Zero-shot performance.** We present the zero-shot results in Table 4. We follow the protocol outlined in (Ni et al., 2022): We use ViT-B as backbone. We first train our model on Kinetics-400 data with 32 frames and conduct the zero-shot evaluation on two unseen datasets (HMDB51 and UCF101). Compared to VLM-based transfer methods, our approach achieves a comprehensive optimal result, securing the second-best performance on HMDB51 and the best on UCF101. The consistently strong results demonstrate the practicality and effectiveness of our method. We attribute this superiority to the incorporation of diverse textual data

Table 4: Zero-shot results on HMDB51 and UCF101. **Bold** indicates the best result, and underline represents the second-best result.

| Method | HMDB51 | UCF101 |
|---|---|---|
| ActionCLIP (Wang et al., 2021) | 40.8±5.4 | 58.3±3.4 |
| X-CLIP (Ni et al., 2022) | 44.6±5.2 | 72.0±2.3 |
| Vita-CLIP (Wasim et al., 2023) | 48.6±0.6 | 75.0±0.6 |
| DiST (Qing et al., 2023) | 55.4±1.2 | 72.3±0.6 |
| ALT (Chen et al., 2024) | 52.9±1.0 | 79.4±0.9 |
| MoTED (Zhang et al., 2024) | **58.2±1.1** | 78.3±0.6 |
| Ours | 56.5±1.0 | **79.9±0.5** |

and the comprehensive spatio-temporal understanding capabilities, which together alleviate the challenges of adapting to new scenarios.

**Visualization Analysis.** As shown in Fig. 4, we illustrate the effectiveness of semantic and visual enhancement through visualizing the important area by GradCAM (Selvaraju et al., 2017). In the left, we find the UMT (baseline) is confused by the complex background, while our method successfully capture action area: basketball hoop and the player. We attribute such success to the aid of text supervision and spatial enhancement, as the text provide detailed description which would help model to understand the abstract action, while the spatial enhancement helps to capture the critical cues. In the right the UMT (baseline) is dominated by the background area, we attribute

to the difficulty of capturing small area, while our method, thanks to the visual enhancement from image encoder, our method is capable of capturing precise area regardless of their sizes.

## 5 CONCLUSION

In this paper, we propose a cross-modal collaborative knowledge transfer method for video understanding. We introduce an Attentive Spatial Knowledge Transfer method for transferring spatial information from the image-side of VLM. Experiments verify a more precise spatial capture ability after the distillation. On the other hand, we enhance the semantic understanding ability by Contrastive Textual Knowledge Transfer. Experiments show the diverse descriptions could cover the large intra-variance within an action, helps build a fine-grained video feature space. Our method achieves the state-of-the-art results under different settings which validates the effectiveness of our method.

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

# A APPENDIX

## A.1 ABLATION STUDY ON KINETICS-400.

**Contribution of each component.** We conduct ablation experiments on Kinetics-400 to verify the effectiveness of each component. We utilize UMT-BLi et al. (2023a) as our baseline model trained with 16 frames and only $\mathcal{L}_{ce}$.

As shown in Table 5, incorporating the cross-modal contrastive loss $\mathcal{L}_{text}$ results in 85.97% Top-1 accuracy, surpassing the baseline by 1.15%, demonstrating the effectiveness of textual knowledge transfer in enforcing fine-grained text-video alignment. Additionally, we achieve the highest result of 86.30% Top-1 accuracy by further adding the feature distillation loss $\mathcal{L}_{image}$, suggesting that general spatial knowledge from VLM enhances video adaptation.

Table 5: Contribution of each component on Kinetics-400.

| $\mathcal{L}_{ce}$ | $\mathcal{L}_{text}$ | $\mathcal{L}_{image}$ | Top-1 |
|---|---|---|---|
| ✓ | | | 84.82 |
| ✓ | ✓ | | 85.97 |
| ✓ | ✓ | ✓ | 86.30 |

**Class-level performance analysis.** As our method boosts performance across over 100 classes, we highlight in Fig. 5 the classes where accuracy improvements exceed 10% compared to the baseline. We observe that our method significantly enhances the performance of classes requiring precise understanding of small objects, such as '*drinking beer*', '*dunking basketball*', and '*applying cream*', demonstrating the effectiveness of distilling video features from the image encoder of VLM. Additionally, classes with large intra-variance, such as '*slapping*' and '*somersaulting*', also show substantial improvements, showcasing the efficacy of the textual contrastive loss.

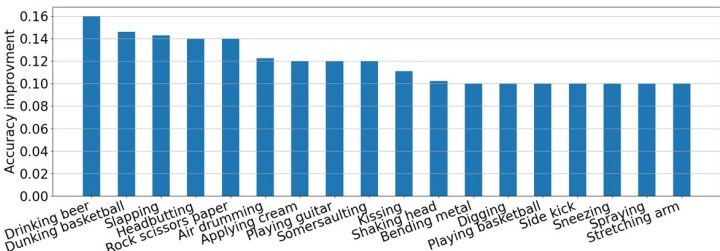

Figure 5: Class-level improvements on Kinetics-400.

## A.2 DETAILS OF DECOMPOSE-EXPAND PROMPT METHOD.

**Prompts used for GPT-4o.** We first extract the key words for each class. Given a large language model (*e.g.*, GPT-4o), we query it with the following prompt:

```
Here is the action list [action name1, action name2,...].what are
the most identifying visual characteristics such as object,body
parts to distinguish them?
```

Next, we ask GPT-4o to generate the diverse descriptions based on these keywords by using the following prompt:

```
Here is the identifying visual characteristics for the action
[action name]: keyword1, keyword2, keyword3. Please describe the
action in different sentences by using the keywords above.
```

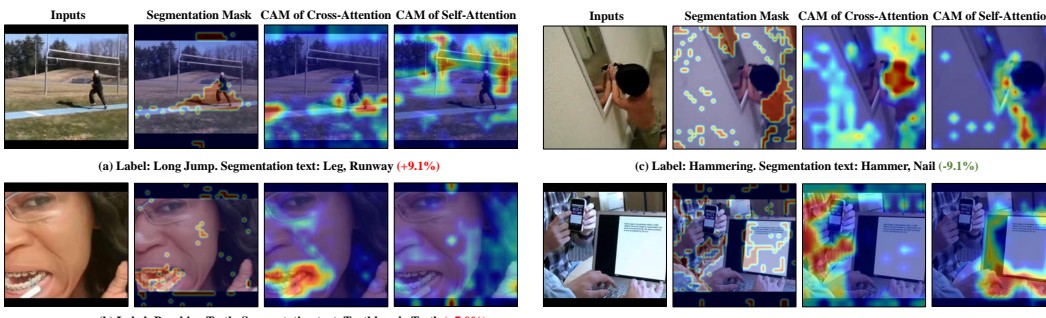

(a) Label: Long Jump. Segmentation text: Leg, Runway (+9.1%)

(c) Label: Hammering. Segmentation text: Hammer, Nail (-9.1%)

(b) Label: Brushing Teeth. Segmentation text: Toothbrush, Teeth (+7.0%)

(d) Label: Typing. Segmentation text: Screen, Keyboard (-7.0%)

Figure 6: Left: Visualization of the classes that cross-attention methods outperforms the self-attention method. Red number indicates the relative performance **gain**. Right:Visualization of the classes that the cross-attention method underperforms the self-attention method. Green number indicates the relative performance **drop**.

**Example of descriptions generated by the prompts** In Table 6, we provide several class descriptions. It is evident that our descriptions encompass more details and demonstrate greater semantic diversity.

Table 6: Examples of generated descriptions on Kinetics-400.

| Action | Descriptions |
|---|---|
| Applying Cream | Squeezing lotion from a bottle and applying it to the skin. |
| | Moisturizing skin by applying lotion from a bottle. |
| | Gently rubbing cream onto the skin using a lotion bottle. |
| Slapping | Delivering a sharp slap with hand. |
| | Hand swinging in motion of slap. |
| | Hand delivering a stinging slap. |

## A.3 EXPLORATION OF THE PERFORMANCE DISCREPANCY BETWEEN TWO GATE METHODS.

In the ablation study of the gate mechanism, we observe that the cross-attention method performs slightly worse than the self-attention method. To investigate the reasons behind this discrepancy, we compare the CAMs of the two attention mechanisms on four test videos. The results are presented in Fig. 6.

As shown in Fig. 6, segmentation quality plays a crucial role in learning comprehensive representation. When the segmentation mask successfully captures the region corresponding to the keyword, it guides the model to highlight attention on the correct areas. In contrast, the self-attention method tends to misinterpret unrelated regions. However, when the segmentation mask fails to accurately capture the keyword's region, the model is misled by the erroneous segmentation, resulting in incorrect predictions.

Unfortunately, we observe that segmentation results are more likely to become unstable or disorganized when the objects of interest are either too small or obscured, , while the self-attention method generally demonstrates robust spatial representation learning.

## A.4 VISUALIZATION OF THE SPATIAL UNDERSTANDING CAPABILITIES OF DIFFERENT MODELS

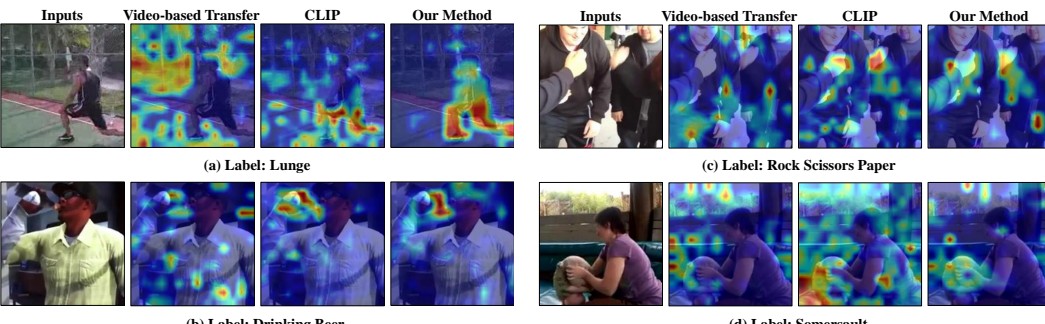

Figure 7: Illustration of spatial understanding capabilities of different models.

We provide supplementary examples in Fig. 7 to further illustrate the spatial understanding capabilities of video pre-trained models, VLMs (e.g., CLIP), and our proposed method. These examples demonstrate that CLIP exhibits superior spatial understanding, excelling at focusing on foreground objects compared to video pre-trained models, which is consistent the conclusion in Fig. 1(c). This highlights the rationale for leveraging VLMs to enhance the spatial understanding of video pre-trained models. The results from our method (the last column) further validate this motivation.

## A.5 TRAINING CONFIGURATIONS

In Table 7, we present the detailed training configurations used in the experiments of the main manuscript. Notably, we exclude the strong augmentations—Mixup (Zhang et al., 2017), Cut-Mix (Yun et al., 2019), and label smoothing (Müller et al., 2019)—as these techniques inevitably alter the semantics of each sample. This allows us to more clearly evaluate the performance of our method.

Table 7: The training configurations on Kinetics-400 and UCF101.

| Settings | Kinetics-400 | UCF101 |
|---|---|---|
| *Optimization* | | |
| Optimizer | AdamW | AdamW |
| Optimizer betas | (0.9,0.999) | (0.9,0.999) |
| Batch size | 192 | 48 |
| Lr. schedule | cosine decay | cosine decay |
| Warmup schedule | linear | linear |
| Base lr | 1.5e-4 | 9.4e-5 |
| Minimal lr | 7.5e-7 | 1.9e-7 |
| Layer-wise lr decay | 0.75 | 0.75 |
| Weight decay | 0.05 | 0.05 |
| Epochs | 40 | 40 |
| *Data augmentations* | | |
| flip augmentation | yes | yes |
| repeated augmentation | 2 | 2 |
| label smoothing | 0.0 | 0.0 |
| Mixup | 0.0 | 0.0 |
| Cutmix | 0.0 | 0.0 |

