# OpenReview forum: "Enhancing Video Understanding with Vision and Language Collaboration"
_ICLR.cc/2025/Conference — Submitted to ICLR 2025_

### Official Review · Reviewer_FYGm · 2024-10-31

**Soundness:** 2
**Presentation:** 2
**Contribution:** 2
**Rating:** 3
**Confidence:** 4

**Summary:**

This paper proposes a cross-modal collaborative knowledge transfer method for video action recognition. The framework consists of three key components: video adaptation, attentive spatial knowledge transfer and contrastive textual knowledge transfer. These three components correspond to three loss functions. Experiments on three datasets achieve state-of-the-art results.

**Strengths:**

+ Introducing VLMs and transfer learning for video understanding is reasonable.

+ The experimental results show slight improvements over previous methods.

**Weaknesses:**

- The paper conducts experiments on video action recognition, but the title and abstract do not explicitly mention this task. Video understanding is a relatively broad concept.

- The three technical contributions result in only marginal improvements, especially as shown in Table 3. It is unclear which loss function serves as the main loss.

**Questions:**

-  Why does the learning rate of the optimizer reported in the implementation details section differ from that in the appendix?

-  Why were experiments not conducted on the mainstream video dataset Something-Something V2 (SSV2) for action recognition?

-  How are the weight coefficients for the loss functions obtained?

-  Line 72 should be written as (e.g., CLIP). There are many similar issues throughout the manuscript.

-  Many equations are missing symbols, such as Equations (6) and (7). Additionally, the bold and italic formatting in the equations is inconsistent.

1. Line 238 is missing punctuation.

2. Line 313, 'Eq. equation 2' is so poor.

3. The capitalization of the first letters in the paper title is inconsistent.

---

> ### Author Response · Authors · 2024-11-27
>
> W1: Unmatched title: Thanks for your helpful suggestions. The experiments have been extensively conducted on popular action recognition datasets, and we have revised our title accordingly for improved clarity.
>
> W2: marginal improvements: Tab. 3 displays the comprehensive ablation studies by removing or adding the proposed losses, with using cross entropy alone as baseline. The absence of either $\mathcal L_{text}$ or $\mathcal L_{image}$ leads to a performance drop (Tab. 3). In contrast, the table blow shows that incorporating both loss terms enables our method to surpass the baseline by 1.56% and 1.48% on UCF101 and Kinetics-400, respectively, demonstrating that $L_{text}$ and $L_{image}$ provide orthogonal and substantial performance gains.
> | Dataset        |$\mathcal L_{ce}$| $\mathcal L_{text}$|$\mathcal L_{image}$ |Acc (%)  |
> |-----------------|----------|----------|----------|----------|
> | UCF101    |√||      | 95.37    |
> | UCF101    |√|√|  | 96.56    |
> | UCF101    |√|√|√| 96.93    |
> | Kinetics-400    |√|| | 84.82    |
> | Kinetics-400 |√|√|| 85.97   |
> | Kinetics-400   |√|√|√| 86.30 |
>
> Q1, Q4, Q5:
> Sorry for the confusion. We have corrected these mistakes in the manuscript.
>
> Q2: Experiments on SSV2.
> The table below presents the comparison on SSV2, including recent video-based and VLM-based transfer methods using the same ViT-L backbone. Our method achieves 74.9% Top-1 accuracy and 94.8% Top-5 accuracy. Notably, SSV2 places greater emphasis on temporal understanding compared to Kinetics-400, which explains why video-based transfer methods (e.g., VideoMAE) typically outperform VLM-based approaches. By leveraging a video pre-trained model to retain temporal understanding while integrating knowledge from VLM, our method achieves state-of-the-art performance.
>
> | Method       | Publication | Backbone | Inputs | Top-1 | Top-5 |
> |--------------|-------------|----------|--------|-------|-------|
> | VideoMAE[1]     | NeurIPS2022 | ViT-L    |   $16\times224^{2}$     | 74.3  | 94.6  |
> | DiST[2]         | ICCV2023    | ViT-L    |    $16\times224^{2}$    | 73.1  | 93.2  |
> | MoTED[3]        | CVPR2024    | ViT-L    |     $16\times224^{2}$   | 73.8  | 93.8  |
> | Qian et al.[4]  | ECCV2024    | ViT-L    |   $32\times224^{2}$     | 73.6  | 94.3  |
> | Ours         |             | ViT-L    |   $16\times224^{2}$      | 74.9  | 94.8  |
>
> [1] VideoMAE: Masked Autoencoders are Data-Efficient Learners for Self-Supervised Video Pre-Training, NeurIPS2022.
>
> [2] Disentangling Spatial and Temporal Learning for Efficient Image-to-Video Transfer Learning, ICCV2023.
>
> [3] Enhanced Motion-Text Alignment for Image-to-Video Transfer Learning, CVPR2024.
>
> [4] Rethinking Image-to-Video Adaptation: An Object-centric Perspective, ECCV2024.
>
> Q3: Ablation study on weight coefficients for the loss function
> We conduct grid search on the weight coefficients to determine the optimal values. The results in the table below demonstrate that our method is robust to variations in the two hyperparameters. For all experiments, we select $λ_{t}=1$ for $\mathcal L_{text}$ and  $λ_{i}=10$ for $\mathcal L_{image}$.
> |   $λ_{t}$  \ $λ_{i} $  | 1      | 5      | 10     | 20     | 100    |
> |-------|--------|--------|--------|--------|--------|
> | 0.1   | 96.09  | 96.38  | 96.38  | 96.01  | 95.55  |
> | 0.5   | 96.14  | 95.92  | 96.28  | 96.14  | 96.11  |
> | 1     | 96.14  | 96.62  | 96.93  | 96.62  | 96.60  |
> | 2     | 96.62  | 96.59  | 96.87  | 96.83  | 96.46  |
> | 10    | 96.83  | 96.44  | 96.55  | 96.44  | 96.32  |

---

### Official Review · Reviewer_fw3o · 2024-11-01

**Soundness:** 3
**Presentation:** 3
**Contribution:** 3
**Rating:** 6
**Confidence:** 4

**Summary:**

This paper proposes to leverage Vision Language Models (VLM) for cross-modal collaborative knowledge transfer to improve the spatial and semantic understanding for video data. The proposed approach aim to reduce the temporal gap between pre-training dataset and downstream tasks.

**Strengths:**

+ This work proposes a cross-modal collaborative knowledge transfer to adapt video pre-trained model for downstream tasks. Specifically, spatial knowledge is distill from VLM's image encoder via attentive spatial knowledge transfer. Then, textual knowledge transfer improve video representation via fine-grained text-video alignment.
+ A gating mechanism is proposed to guide the distillation process to focus more on the action relation region and less on broader content.
+ To improve the text-video alignment, a decompose-expand prompting method is proposed to improve the fine-grained and diverse description. This provide more training cue than a single description or class name. Then, a contrastive textual knowledge transfer learn the semantic knowledge.
+ The proposed method achieve consistent improvement over multiple baselines and datasets.

**Weaknesses:**

I don't see obvious weaknesses in this work. Please refer to the Questions Section for some clarification.

**Questions:**

- On the ablation of negative text sampling, can the authors provide higher number of k (why is 1200 the current limit). In addition, It is no meaningful to stated that sampling 200 negative text yield the third-best Top-1 accuracy. Please provide a better explanation of the initial accuracy drop when k increase from 200 to 600, and subsequently improve again. How does the authors know the diminish was due to easy negatives? Is there a way to measure if easy negative was sampled in early stage?
- On the evaluation of gate mechanism, the improve with gate mechanism is relatively low (+0.24%). It would be to provide analysis on the class that is accurately classified with gate mechanism, and analyse if the corresponding attention map (with and without gate mechanism) match the hypothesis.
- What is the number of description generated for each category. Has the authors analyse the score of the positive text and observed cases where the scores are generally too low for some cases.

---

> ### Author Response · Authors · 2024-11-27
>
> Q1: Ablation study on number of negative descriptions, i.e., k.: Sorry for the any confusion caused by Fig. 3. To fully investigate the irregular fluctuation especially when k ranges from [200,1000], we conducted experiments with varying k using different random seeds. Results of k ranges from [200,1300] are presented in Fig.3. The box plot reveals slight fluctuations when k $<$1000 and a noticeable increase as k$ >$ 1000, which can be attributed to the inclusion of more hard negative descriptions. Based on these results, we selected the optimal k $=$ 1200 for all experiments. Fig. 3 in the manuscript has been updated accordingly.
>
> Q2: Visualization to validate the effectiveness of gate mechanism: We include additional visualization results provided in Fig. 4. These results show that training without the gate mechanism leads to diffused attention, while incorporating it focuses attention more effectively on the foreground, thereby enhancing overall performance.
>
> Q3: Number of the generated descriptions and working principle for positive text: Our description candidates for each class includes 3 kinds of descriptions: 10 different descriptions generated by decompose-expand prompting method, 2 different detailed descriptions generated by GPT 4-o and the original action name. For each sample, we choose the description from the ground-truth description candidates with the highest image-text similarity score as positive. This strategy ensures that the selected positive description has a similarity score equal to or higher than that of the original action name, thus making the positive description is conceptually matched to the sample.

---

### Official Review · Reviewer_paC5 · 2024-11-02

**Soundness:** 3
**Presentation:** 3
**Contribution:** 2
**Rating:** 5
**Confidence:** 4

**Summary:**

This work addresses the limitations of current video pre-trained models by integrating large VLMs into a cross-modal collaborative knowledge transfer framework. This method includes an attentive spatial knowledge transfer method that enhances spatial understanding and a contrastive textual knowledge transfer method for improved video-text alignment.

**Strengths:**

+ The writing is very good, relatively easy to understand, and easy to follow as well.
+ Using VLM to enhance video understanding is a very meaningful issue, and the results of this paper are good.

**Weaknesses:**

- It is insufficient to prove the core motivation using only one figure (Fig. 1 c).
- the article lacks an analysis of the consumption of training costs.

**Questions:**

- In lines 42-44, the author points out the "emphasis on capturing temporal information" and illustrates how this is a common phenomenon through the heatmap in Fig. 1(c). How does this demonstrate that the method proposed in this paper can address this issue, and what is the specific principle behind the solution?
- The description of the experimental results is quite imprecise. The performance of existing methods for Top-1 has exceeded 88.5% [1] and reached 89.7% two years ago [2], while the proposed method only achieves 86.9%. Therefore, it cannot be claimed that this method has achieved SOTA.
- In Line 267, how can we ensure that the descriptions generated by the decompose-expand prompting are accurate? What would be the consequences if there are errors?
- Additionally, please include the Top-5 results in the experimental outcomes.


[1] Rethinking Image-to-Video Adaptation: An Object-centric Perspective
[2] Disentangling Spatial and Temporal Learning for Efficient Image-to-Video Transfer Learning

---

> ### Author Response · Authors · 2024-11-27
>
> W1: Insufficient experiments prove the core motivation in Fig. 1(c): We add additional visualization results in Appendix A.4 to further demonstrate the general spatial understanding capabilities of Vision-Language Models (VLMs), such as CLIP, which excel at focusing on foreground objects. Based on these observations, it is reasonable to leverage VLMs to enhance the spatial understanding capabilities of video pre-trained models.
>
> W2: The table below compares our method with UMT(baseline) in terms of training time on Kinetics-400. All experiments are conducted on the same GPUs using identical training configurations, including batch size, training epochs, and learning rate. Despite comparable training hours, our approach achieves a significant improvement in accuracy.
> | Method      | Backbone | Inputs | Training Hours (h) | Accuracy (%) |
> |-------------|----------|-------|--------------------|--------------|
> | Baseline    | ViT-B    | $16\times224^{2}$    | 48.8               | 84.82        |
> | Our Method  | ViT-B    |  $16\times224^{2}$   | 50.7               | 86.30        |
>
> Q1: Relations between lines42-44 and Fig1(c): Sorry for the confusion. We would like to emphasize that video pre-trained models often lack spatial understanding due to their video-centric pre-training objectives. Specifically, as shown in Fig. 1(c), the video pre-trained model is noticeably distracted by the background, whereas CLIP focuses more on the foreground person. This observation motivates us to leverage CLIP's strong spatial encoding capabilities to enhance the spatial understanding of video pre-trained models, thereby improving video understanding.
>
> Q2 and Q4: Comparison with the state-of-the-art on Kinetics-400:
> Tab. 1 presents a comparison of all methods using ViT-B as the backbone, whereas the works mentioned by the reviewer employ ViT-L to achieve 88.5% [1] and 89.7% [2]. For a fair comparison, we provide an additional experiment using ViT-L in the table below. The results demonstrate that our method achieves the highest performance in both top-1 and top-5 accuracy, surpassing the second-best method, DiST, by 1.0% in top-1 accuracy. In summary, our method outperforms others with both ViT-B and ViT-L as backbones.
> | Method       | Publication | Backbone | Inputs | Top-1 | Top-5 |
> |--------------|-------------|----------|--------|-------|-------|
> | Qian et al. [1] | ECCV2024  | ViT-L    |  $32\times336^{2}$         | 88.5  | 98.0  |
> | DiST [2]     | ICCV2023    | ViT-L    |      $32\times224^{2}$   | 89.7  | 98.5  |
> | Ours         |             | ViT-L    |   $32\times224^{2}$     | 90.7  | 98.6  |
>
> [1] Rethinking Image-to-Video Adaptation: An Object-centric Perspective, ECCV2024.
>
> [2]Disentangling Spatial and Temporal Learning for Efficient Image-to-Video Transfer Learning, ICCV2023.
>
> Q3: Correct usage of generated descriptions by the decompose-expand prompting: To select the conceptually correct positive description, we consider the union of generated descriptions and the original action names of the ground-truth label as text candidates. From these, we choose the one with the highest image-text similarity score as positive. This strategy ensures that the selected positive description has a similarity score equal to or higher than that of the original action name. For creating negative descriptions, we select the bottom-k candidates with the lowest similarity scores, excluding the ground-truth descriptions. This ensures that the selected negative descriptions are irrelevant to the training sample. We have updated the manuscript to clarify this process.

---

### Official Review · Reviewer_Y5QN · 2024-11-04

**Soundness:** 2
**Presentation:** 3
**Contribution:** 2
**Rating:** 5
**Confidence:** 4

**Summary:**

The paper aims to improve the spatial understanding of the existing video-language model for the classification task. This is implemented with three different optimization losses, including the classification task (classify the video into predefined classes), contrastive loss (aligning the video with the text descriptions associated with the video) and spatial knowledge transfer task. The spatial knowledge transfer task utilizes the frozen large vision language model to guide the learning of video encoder, by aligning the video token with the image token. Extensive experiments and ablation are conducted on different video datasets and backbones to demonstrate the effectiveness of the proposed training strategies.

**Strengths:**

1.	Figure 2 is well designed and fully captured the overall training pipeline.
2.	The paper is well written and simple to follow.
3.	The motivation of this work is strong and interesting.

**Weaknesses:**

1.	The main contribution of this work is unclear. The classification loss and contrastive loss are common in video pretraining and video classification. Moreover, the alignment with L1 loss between the image token and the video token is also not considered novel.
2.	The most interesting section in this work is the cross attention section in Eq 6-8, which leverages the segmentation model to guide the alignment. However, while the author shows that the result is not performing well, the explanation in L258-260 is not sound and solid. It will be interesting if the author can provide more study and in-depth discussion.
3.	L297-L299 is unclear. What does it mean by “dissimilar description would be less linked to the action”?
4.	The conclusion from the negative test sampling ablation is unclear. Figure 3 shows a reflection point where the performance of the model has a minimum. Is there any reason why this is happening? What is the variance of each data point in Figure 3?

**Questions:**

While the motivation of this work is strong and interesting, not much novelty is observed in the study. The gain of this work is also minor compared to the previous work and the variance is not reported. Furthermore, some of the ablation study (like the number of negative text) seems to be very sensitive to the variance. It is unclear whether the gain in Table 1 and the ablation in Table 3 are due to the variance.

---

> ### Author Response · Authors · 2024-11-27
>
> W1: unclear contribution: Our main contributions stand in enhancing video understanding by equipping pre-trained models with Vision-Language Models (VLMs), which is under-explored in the literature. However, adapting the visual and textual knowledge of VLMs to video tasks is non-trivial due to modality gaps. To the end, we propose attentive spatial and contrastive textual knowledge transfer modules to unveil the VLM’s power. These modules, combined with simple yet effective loss functions, are extensively validated through ablation experiments, as shown in Tab. 3.
>
> W2: In-depth discussion of the cross-attention method: To explore the cause of the suboptimal performance of the cross-attention method, additional experiments are conducted in Appendix A.3. Specifically, we visualize the segmentation mask and Grad-CAM of a few typical examples in Fig. 6. It reveals the suboptimal performance of the Cross-Attention method arises from unstable segmentation results, particularly when the objects are either too small or obscured.
>
> W3: Unclear description: Thanks for your helpful suggestions. We rely on text-image similarity scores to determine whether descriptions are positive or negative. High scores indicate a strong correlation between the video/frames and the text descriptions, while low scores suggest that the descriptions do not accurately reflect the video content and thus serve as negatives. We have updated it in the manuscript.
>
> W4: Sensitivity analysis of hyperparameter k: Sorry for the any confusion caused by Fig. 3. To evaluate the sensitivity to the number of negative descriptions, namely k, we conducte experiments with varying k values using different random seeds. The box plot reveals slight fluctuations when k $<$ 1000 and a noticeable increase as k $>$ 1000, which can be attributed to the inclusion of more hard negative descriptions. Based on these results, we select the optimal k $=$ 1200 for all experiments. Fig. 3 in the manuscript has been updated accordingly.
>
> Q1: The performance gain: We thank the reviewer for recognizing the motivation behind our work. Our method has been extensively evaluated on three video recognition datasets: UCF-101, Kinetics-400, and Something-Something-v2 (SSv2), achieving consistent improvements. Specifically, on UCF-101, we achieve a notable 1.56\% gain in Tab. 2(b) compared to the strong UMT baseline using ViT-B. On the challenging Kinetics-400 and SSv2 datasets, we obtain improvements of 1.0\% and 0.6\% by using ViT-L, respectively. Results are shown below.
> | Method| Publication |  |   SSV2      |           |  |    Kinetics-400       |           |
> |:-:|:-:|:-:|:-:|:-:|:-:|:-:|:-:|
> |             |             | Inputs       | Top-1     | Top-5     | Inputs | Top-1     | Top-5     |
> | VideoMAE[1] | NeurIPS2022 |   $16\times224^{2}$           | 74.3      | 94.6      |   $32\times320^{2}$    | 86.1      | 97.3      |
> | DiST[2]     | ICCV2023    |    $16\times224^{2}$          | 73.1      | 93.2      |   $32\times336^{2}$    | 89.7      | 98.5      |
> | MoTED[3]    | CVPR2024    |     $16\times224^{2}$         | 73.8      | 93.8      |    $32\times224^{2}$   | 88.8      | 98.2      |
> | Qian et al.[4] | ECCV2024 |     $32\times224^{2}$         | 73.6      | 94.3      |   $32\times224^{2}$    | 88.5      | 98.0      |
> | Ours        |             |        $16\times224^{2}$      | 74.9      | 94.8      |   $32\times224^{2}$    | 90.7      | 98.6      |
>
> [1] VideoMAE: Masked Autoencoders are Data-Efficient Learners for Self-Supervised Video Pre-Training, NeurIPS2022.
>
> [2] Disentangling Spatial and Temporal Learning for Efficient Image-to-Video Transfer Learning, ICCV2023.
>
> [3] Enhanced Motion-Text Alignment for Image-to-Video Transfer Learning, CVPR2024.
>
> [4] Rethinking Image-to-Video Adaptation: An Object-centric Perspective, ECCV2024.
>
> Q2: Ablation study in Tab. 3: In Tab. 3, we present comprehensive ablation studies on the proposed loss functions $\mathcal L_{text}$ and $\mathcal L_{image}$. Notably, these two loss functions provide orthogonal performance gains. Different configurations of these loss functions consistently enhance the baseline performance. Further validation is provided by the ablation study on Kinetics-400 in Tab. 5, which demonstrates that introducing the proposed loss functions yields the best performance.

---

### Author Response · Authors · 2024-11-27

We appreciate the detailed and constructive feedback from all the reviewers. Specifically, we are pleased that they highly value our motivation (all the reviewer) as well as the clarity and quality of the writing of the paper(Y5QN,  paC5). Before addressing the individual reviews, we briefly outline the manuscript revisions and recurring points highlighted in the feedback.

1. Extended Evaluation:

   $\circ$ More datasets with larger backbone: We conduct experiments on Kinetics-400 and SSV2 using ViT-L as backbone.

   $\circ$ Training cost: We add comparison of the training hours between baseline and our method.

2. Expanded Ablation Study:

   $\circ$ Number of negative texts: we rerun the experiments with multiple random seeds to obtain more stable results with a new analysis.

   $\circ$ Weigh coefficient: we add grid search result on weight coefficient.

   $\circ$ Gate mechanism: We add qualitive analysis on the self-attention method in Fig. 4 and in-depth discussion of the cross-attention method in Fig .6.

3. Enhanced writing

   $\circ$ We revise the manuscript to address any confusing descriptions based on the reviewers' comments, including a more rigorous introduction of our motivation, a detailed explanation of the selection process for positive and negative texts, a comprehensive description of punctuation usage, and so on. Additionally, grammar issues have been corrected.

---

> ### Author Response · Authors · 2024-12-01
>
> We sincerely thank all the reviewers for their insightful comments, which have significantly enhanced the quality of our manuscript. We look forward to receiving further feedback and are eager to engage in discussions to address any concerns to the best of our ability.

---

### Meta-Review · Area_Chair_FNUW · 2024-12-17

**Metareview:**

This paper presents an algorithm that applies general methodologies of vision-language models to enhance video understanding, in particular, video action recognition. There are two main technical improvements, a knowledge distillation module and a fine-grained VL alignment module. Experiments are performed on traditional video understanding benchmarks, such as UCF101, HMDB51, and Kinetics400 and show improvements.

This paper receives an overall rating of 3/5/5/6, falling below the acceptance threshold. The reviewers raised concerns about the novelty of the paper. The AC reads the paper and finds the studied problem somewhat outdated, e.g. the studied datasets are old benchmarks released some 10 years ago, and currently, video understanding tasks have evolved from closed-domain to open-domain, but the paper does not consider this new trend. In particular, it adapts a large pre-trained model (that has an open-domain recognition ability) to the restricted domain, limiting the value of the paper and its interest to the community. The AC finds no reason to overturn the reviewers' recommendation.

**Additional Comments On Reviewer Discussion:**

Only one reviewer responded after the rebuttal, insisting on borderline rejection. The reviewers were mainly concerned about the novelty of the paper. The paper is well-written, but some aspects are not sufficiently clear, raising concerns from the reviewers (although the authors tried to provide further information in the rebuttal).

---

### Decision · Program_Chairs · 2025-01-22

Reject